# Environmental Factors, More Than Spatial Distance, Explain Community Structure of Soil Ammonia-Oxidizers in Wetlands on the Qinghai–Tibetan Plateau

**DOI:** 10.3390/microorganisms8060933

**Published:** 2020-06-21

**Authors:** Wen Zhou, Xiaoliang Jiang, Jian Ouyang, Bei Lu, Wenzhi Liu, Guihua Liu

**Affiliations:** 1CAS Key Laboratory of Aquatic Botany and Watershed Ecology, Wuhan Botanical Garden, Chinese Academy of Sciences, Wuhan 430074, China; zhouwen@wbgcas.cn (W.Z.); jiangxl@wbgcas.cn (X.J.); ouyangjian07@foxmail.com (J.O.); lubei@wbgcas.cn (B.L.); 2Hubei Key Laboratory of Wetland Evolution & Ecological Restoration, Wuhan Botanical Garden, Chinese Academy of Sciences, Wuhan 430074, China; 3College of Life Science, University of Chinese Academy of Sciences, Beijing 100049, China; 4Research Center for Ecology and Environment of Qinghai–Tibetan Plateau, Tibet University, Lhasa 850000, China; 5College of Science, Tibet University, Lhasa 850000, China; 6Center for Plant Ecology, Core Botanical Gardens, Chinese Academy of Sciences, Wuhan 430074, China

**Keywords:** nitrification, the Qinghai–Tibet Plateau, high-elevation wetland, microbial community structure

## Abstract

In wetland ecosystems, ammonia oxidation highly depends on the activity of ammonia-oxidizing archaea (AOA) and ammonia-oxidizing bacteria (AOB), which are, therefore, important for studying nitrogen cycling. However, the ammonia-oxidizer communities in the typical high-elevation wetlands are poorly understood. Here, we examined ammonia-oxidizer communities in soils from three wetland types and 31 wetland sites across the Qinghai–Tibetan Plateau. The *amoA* gene of AOA and AOB was widespread across all wetland types. *Nitrososphaera* clade (Group I.1b) overwhelmingly dominated in AOA community (90.36%), while *Nitrosospira* was the principal AOB type (64.96%). The average abundances of AOA and AOB were 2.63 × 10^4^ copies g^−1^ and 9.73 × 10^3^ copies g^−1^. The abundance of AOA *amoA* gene was higher in riverine and lacustrine wetlands, while AOB *amoA* gene dominated in palustrine wetlands. The environmental conditions, but not spatial distance, have a dominant role in shaping the pattern of ammonia-oxidizer communities. The AOA community composition was influenced by mean annual temperature (MAT) and mean annual precipitation (MAP), while MAT, conductivity and plant richness, pH, and TN influenced the AOB community composition. The net nitrification rate had a significant correlation to AOB, but not AOA abundance. Our results suggest a dominant role for climate factors (MAT and MAP) in shaping community composition across a wide variety of wetland sites and conditions.

## 1. Introduction

Nitrification is the biological oxidation of ammonia into nitrate [1] and it plays essential roles in the wetland nitrogen cycle and N_2_O production. Ammonium oxidation, as the first and rate-limiting step of nitrification, was mainly catalyzed by ammonia-oxidizing bacteria and ammonia-oxidizing archaea [2,3]. Both groups of taxa perform a similar function of ammonia monooxygenase enzymatic pathway; therefore, the *amoA* gene is commonly used as a functional marker for ecological assessment of these taxa [2,4]. However, many previous studies suggested that their relative abundances vary from fractional to several orders of magnitude, depending upon a variety of environmental factors—such as pH, salinity, organic carbon, temperature, ammonium and moisture content [5,6,7,8,9,10]. For instance, AOA dominates nitrification in low ammonium, low pH, and high temperature environments, while AOB dominates nitrification in high ammonium and low temperature environments [5,11,12,13,14].

While many studies have demonstrated that AOA and AOB are widely distributed and regulated by the environment factors [15], less is known regarding the ammonia-oxidizer community in high-elevation wetland ecosystems. We do not know whether high-elevation wetlands have microbial aspects that are similar to other wetlands or if their microbial community structure and function are unique. Two previous studies have investigated ammonia-oxidizer communities in the overlying water and sediments of five rivers in the Qinghai–Tibetan Plateau, and found that the high-elevation conditions (low temperature, low ammonium concentration, and intensive solar radiation) shaped distinctive community compositions and distribution patterns for ammonia oxidizers in the five rivers, as compared with low-elevation rivers [10,16]. These observations highlight the necessity of better understanding how a wider range environmental condition can regulate ammonia-oxidizer communities. However, it is unknown whether these findings are applicable to other high-elevation wetlands.

The wetland on the Qinghai–Tibetan Plateau is the largest high-elevation wetland region on earth, holding over 131,894 km^2^ of area and accounting for approximately 20% of wetlands in China [17]. There are various types of wetland ecosystems in the Qinghai–Tibetan plateau, such as riverine, lacustrine, and palustrine wetlands, according to hydrologic connection, soil moisture, and vegetation [18]. These wetlands are the major nitrogen sinking pools and they have a high sensitivity of environmental changes [19]. Plateau wetlands typically show a sharp decrease of temperature and nitrogenous compounds, but an intensive increase of solar radiation, due to its unique high-elevation conditions and the low human disturbance.

In the present study, we investigated 31 natural wetlands from three wetland types on the Qinghai–Tibetan Plateau. For these 31 wetlands, the average elevation is higher than 4000 m above sea level, and the distance between sites reached upwards of 2457 km. Moreover, the environmental settings were highly heterogeneous. For example, the soil conductivity ranged from 61 to 2848 µS cm^−1^, and the average temperature from −3.52 °C to 17.62 °C. Given these novel environmental characteristics, we reported the abundance, composition, and diversity of ammonia-oxidizer communities in wetland sediments, and examined their correlations with geographic distance and potential environmental factors. Specific questions we address include: (1) what are the compositions and relative abundances of microbial communities in alpine wetland sediments? (2) How do microbial communities vary across geographic distance and environmental gradients? And (3) what are the key drivers if such variations exist? 

## 2. Materials and Methods

### 2.1. Study Area and Sampling

The elevation of our study area ranges from 254 to 5151 m and the latitude lies between 28.16 and 37.46° (N). A total of 31 isolated wetlands situated in both Qinghai Province and the Tibetan autonomous Region of China were selected, covered a wide range stretching 1034 km from north to south and 2457 km from east to west (Figure 1). These wetlands are in varied hydrological and physicochemical environments, encompassing 18 lacustrine (lake margin), seven riverine, and six palustrine wetlands, respectively (Appendix A), based on the definition developed by Tammi [20]. Surface soils (0–5 cm) of each wetland site were sampled with a 5-cm-diameter hand corer. Each composite soil sample was randomly collected from five selected locations in a 1 m^2^ plot. After being passed through a 2 mm sieve to remove the plant residues, root fragments, and gravel, these five soil samples were mixed and homogenized, and were then subdivided into two subsamples. One was stored at approximately 5 °C in a portable refrigerator to determine the soil properties, and the other was stored in liquid nitrogen prior to DNA extraction. Surface water samples (upper 50 cm) were collected in each site for water quality determination. The water samples were stored at approximately 5 °C for property testing.

### 2.2. Vegetation Survey and Measurements of Environmental Factors

At each sampling site, the latitude, longitude, and elevation were recorded. The mean annual temperature and mean annual precipitation of each site were extracted by interpolation [21] from a 1-km resolution climate dataset of the Chinese ecosystem research network (CERN) in ArcGIS 10.0 (ESRI Inc., Redlands, CA, USA). The plant species richness of each site was taken as the total number of recorded vascular plants per 1 m^2^ plot. The plant coverage was visually estimated in each quadrat while using a 1 × 1 m grid frame, which was divided into 10 × 10 cm squares [22].

Water temperature, pH, and electrical conductivity were determined in situ with a Multi-Parameter Water Quality Sonde (YSI 6920, Yellow Springs, OH, USA) at a depth of 50 cm below the water surface. Samples for the total nitrogen (TN), total carbon (TC), and total organic carbon (TOC) measurement were analyzed with a total organic carbon analyzer (Shimadzu TOC-Vc series, Tokyo, Japan) that was equipped with a total nitrogen module. The concentration of total phosphorus (TP) was determined using the colorimetric method with a spectrophotometer (UV-1800, Shimadzu, Tokyo, Japan) after digestion by potassium peroxydisulfate solution.

Soil temperature at 5 cm depth of each wetland site was measured using a thermometer with a stainless probe. In the laboratory, soil pH and electrical conductivity were determined in soil water extract (1:5 w/v soil to water ratio) while using a pH/conductivity meter. Soil moisture content was gravimetrically assessed by drying the soil at 105 °C until constant mass was reached. Soil total carbon (STC) and total nitrogen (STN) were measured with an elemental analyzer (Vario TOC cube, Hanau, Germany). The concentration of soil total phosphorus (STP) was measured by the molybdenum blue method with a spectrophotometer (Shimadzu, Tokyo, Japan) after digestion. The soil ammonium (SNH_4_^+^) and nitrate (SNO_3_^−^) were determined with a continuous flow auto-analyzer (EasyChem Plus, Systea, Italy).

### 2.3. Determination of Soil Nitrification Rate

The soil nitrification rates were determined using the shaken-slurry method, as described by Hart et al. [23]. Specifically, 5 g of fresh soil from each site were weighted into a 250 mL sterile Erlenmeyer flask. 100 mL of a 1 mM phosphate buffer (pH 7.4) and 0.5 mL of a 0.25M (NH_4_)_2_SO_4_ solution were added to allow gas exchange during the nitrification process. All of the flasks were incubated on an orbital shaker with a 180 rpm speed at room temperature for 24 h. 10 mL subsamples were removed from the slurry at 1, 4, 10, 16, and 24 h after the start of incubation, and then filtrated through glass microfiber filters (Whatman, UK) after centrifugation at 3000 rpm for 5 min. The concentrations of NH_4_^+^ and NO_3_^−^ in filtrate were measured by automatic nutrient analyzer (EasyChem plus, Systea, Italy). Because the nitrite concentrations were negligible in the incubations, the potential nitrification rate (PNR) was calculated as the change in NO_3_^−^ concentration per unit time. The determination of net nitrification rate (NNR) was followed the same protocol as PNR, except that the phosphate buffer and (NH_4_)_2_SO_4_ solution were replaced by 100 mL in situ water.

### 2.4. DNA Extraction, Amplification, Cloning and Sequence Analysis for Ammonia-Oxidizing Genes 

Genomic DNA was extracted from approximately 0.2 g of mixed soil subsamples while using a PowerSoil DNA Isolation Kit (MoBio Laboratories, Inc., Carlsbad, CA, USA) according to the manufacturer’s protocol. The DNA concentrations were spectrophotometrically measured with a NanoDrop 2000 (Thermo Fisher Scientific, Waltham, MA, USA). We further confirmed the intact DNA by electrophoresis on a 1% TAE-agarose gel. The primer pairs Arch-amoAF/Arch-amoAR [24] and amoA-1F/amoA-2R [25] were adopted to amplify and construct the clone libraries of AOA- and AOB-*amoA* (ammonia monooxygenase subunit A) genes, respectively. The sequences of primers and thermal cycling procedures are shown in Appendix A. Each reaction was performed in a 25 μL volume consisting of 1 μL of DNA template (10–100 ng/μL), 0.5 μL of each primer (10 mM), 0.2 μL of rTaq polymerase (5 U/μL) (TaKaRa, DaLian, China), 0.5 μL of deoxynucleotide triphosphates (dNTP, 10 mM), and 2.5 μL of 10× buffer. The PCR product was gel-purified and then ligated into the pMD18-T vector (TaKaRa, DaLian, China), which was transformed into Trans-5α competent cells (TransGen Biotech, Beijing, China) and then incubated at 37 °C for about 14 h. The insertion of an appropriate-sized DNA fragment was determined by PCR amplification with the primer set M13F and M13R (Sangon Biotech Co., Ltd., Shanghai, China).

Approximately sixty positive clones with the correct size from each site were selected for sequencing while using ABI-3730XL automated sequencer (Applied Biosystems, Foster City, CA, USA) by Sangon Biotech Co., Ltd. Sequences with poor-quality or insufficient length were discarded by Geneius Pro 8.0.2 software (Biomatters Ltd., Auckland, New Zealand), and other sequences were aligned using MAFFT program [26]. The sequences sharing 97% similarity were grouped into the same operational taxonomic unit (OTU) while using the program Mothur with the furthest neighbor algorithm [27]. Rarefaction analysis was performed by using Mothur to show whether the majority of OTUs was already covered by sampling. The taxonomic identities of the main OTUs (at least containing two clones) of AOA and AOB were determined by constructing phylogenetic trees with reference sequences obtained from the National Centre for Biotechnology Information (NCBI) database while using MEGA v7 [28]. Bootstrap analysis was used to estimate the reliability of phylogenetic reconstructions (1000 replicates). The archaeal and bacterial amoA sequence data reported are available in the GenBank database under the accession numbers MG574595-MG574721 (for archaeal amoA) and MG574722-MG574819 (for bacterial amoA).

### 2.5. Real-Time Quantitative PCR

Copy numbers of the *amoA* gene of ammonia-oxidizing bacteria and archaea were determined in triplicate while using Roche LightCycler480 software version 1.5 with the fluorescent dye SYBR green quantitative PCR method. The Arch-*amoA* and *amoA* gene fragments were amplified using the primer sets Arch-amoAF/Arch-amoAR and amoA-1F/amoA-2R, respectively. qPCR amplification was performed in a 25-μL reaction mixture that consisted of 10 μL of SybrGreen qPCR Master Mix (2×), 1 μL of each primer (10 μM), and 2 μL of template DNA. The primers and qPCR thermal profiles are listed in Appendix A. The positive colonies confirmed that Arch-*amoA* and *amoA* genes were successfully ligated into the pMD18-T vector (TaKaRa, Dalian, China). The plasmid DNA was extracted from the positive colony using the SK8191 SanPrep Kit (Sangon Biotech Co., Ltd., Shanghai, China). The concentration of the plasmid DNA was measured on a Nanodrop 2000 spectrophotometer (Thermo Fisher Scientific, Waltham, USA). Standard curves were constructed with ten-fold serial dilutions of a known amount of plasmid DNA involving the target genes. For all assays, the amplification efficiencies were 89–110% and R^2^ values were all over 0.99.

### 2.6. Definition of Abundant and Rare Taxa

For the definitions of abundant and rare communities, we followed the most-applied definitions in recent literature that combined their local and regional relative abundances [29]. The abundant OTUs were defined as those with a relative abundance of >1% in local samples and a mean relative abundance of >0.1% in all samples. The OTUs that had a mean relative abundance of <0.01% in local samples and the mean relative abundances of <0.001% in all samples were defined as regionally rare OTUs.

### 2.7. Statistical Analyses

Alpha diversity (expressed as OTU richness, evenness and Shannon index) of each sample was calculated in R (3.5.2) with vegan package [30]. Kruskal–Wallis H test was used to compare the difference in gene abundance between ammonia-oxidizing archaea and bacteria, and also be used to estimate the differences in alpha diversity among the three wetland types, because the data failed the normality test. The relationships of geographical distance and sample ordination on taxonomic (Bray–Curtis distance), as well as environmental dissimilarity (Euclidean distance based on 19 environmental variables) were investigated based on the Mantel test with 999 permutations [30]. Non-metric multidimensional scaling (NMDS) ordination was performed based on unweighted Unifrac distances by using GUniFrac and vegan package in order to visualize the similarity of community structure among different three wetlands [31]. The principal coordinates of neighbor matrices (PCNMs) approach was used to calculate a set of spatial variables based on the longitude and latitude coordinates of each sampling site [32].

Although variation partitioning analysis (VPA) is widely used in ecological research to distinguish between environmental constraints and potential dispersal limitation, it is sometimes difficult to correctly predict the environmental and spatial components of community structure, especially in the simulation models researches [33]. Therefore, VPA, Mantel, and partial Mantel test were conducted in this study, in order to evaluate the relative importance of the selective and neutral processes in shaping the ammonia-oxidizer community. We correlated the dissimilarities of community composition with those of environmental factors using the partial Mantel test to identify the environmental drivers in the ammonia-oxidizing microbial communities. Before the analysis, the environmental variables with high variable inflation factor (VIF > 10) were eliminated to limit collinearity among factors. Further, we calculated the proportionate contribution that each environmental variable made to the coefficient of determination (R^2^) in multiple regression by ‘relweight’ function in R [34] in order to clarify the relative contribution of these drivers.

## 3. Results

### 3.1. Physicochemical Properties of the Samples

During sampling periods, soil temperatures (8.8 °C to 29.6 °C) were significantly correlated to elevation (*r* = −0.40, *p* < 0.05, *n* = 31) (Appendix A). Soil conductivity significantly increased from 61 to 2848 µs cm^−1^ with the increase of latitude (*r* = 0.51, *p* < 0.01, *n* = 31). The contents of ammonium (NH_4_^+^) and total carbon ranged from 0.201 to 5.875 mg-N kg^−1^ and from 0.453 to 13.451 mg kg^−1^ dry sediment, respectively. No significant differences in the physicochemical properties were detected among wetland types (*p* > 0.05).

### 3.2. Abundance of Ammonia-Oxidizing Microorganisms (AOM) and Taxonimic Classification

A total of 1265 archaeal *amo*A sequences and 1567 bacterial *amo*A sequences were obtained, which clustered into 127 and 161 OTUs at 97% sequence similarity level for AOA and AOB, respectively. The rarefaction curves showed that the vast majority of AOM taxa in the sediments of wetlands were recovered in our samples (Appendix A).

We observed that the abundance of AOA *amoA* of all soil samples varied between 520 and 2.36 × 10^5^ copies g^−1^ dry soil, with the lowest and the highest values being observed in the lacustrine and riverine wetlands, respectively (Figure 2 and Appendix A). For the AOB *amoA* gene, the abundance ranged from 69.2 to 8.9 × 10^5^ copies g^−1^ dry soil, with the lowest and the highest values being observed in the lacustrine and palustrine wetlands, respectively (Figure 2 and Appendix A). The average AOA abundance level of all samples (2.63 × 10^4^ copies g^−1^ dry soil) was significantly higher than that of AOB (9.73 × 10^3^ copies g^−1^ dry soil) (Kruskal-Wallis, *p* < 0.01, Figure 2). However, in specific wetland type, the average AOB abundance level of palustrine wetlands (3.06 × 10^4^ copies g^−1^ dry soil) were significantly higher than that of AOA (1.36 × 10^4^ copies g^−1^ dry soil).

There were no rare OTUs observed in our study according to the definition of abundant and rare taxa. The patterns of taxonomic compositions were very similar between all and abundant taxa in AOA communities. While as for AOB communities, the percentage of unique OTUs for abundant taxa was much lower in palustrine wetlands (Appendix A).

According to the phylogenetic trees, the dominant (90.36%) of AOA sequences were clustered into the *Nitrososphaera* clade (Group I.1b), followed by *Nitrosopumilus* (Group I.1a, 5.14%). No sequence was affiliated to *Nitrosotalea* (the Group I.1a-associated cluster) (Appendix A). As for AOB, 64.96% of all bacterial amoA sequences were grouped into the *Nitrosospira* cluster, while approximately 24% and 7% of total AOB sequences affiliated with *N. eutropha* and *N. oligotropha*, respectively (Appendix A).

### 3.3. The Nitrification Potential of Samples

The net nitrification rate and potential nitrification rate of soil were 0.10 ± 0.02 and 0.37 ± 0.02 mg N kg^−1^ soil d^−1^, respectively, while there was no significant difference in each types of wetland (Figure 3). NNR and PNR were both significantly correlated with AOB abundance, but not with the abundance of AOA. When the three types of wetland were analyzed separately, the abundance of AOA from palustrine wetlands were found to strongly correlate with PNR (Figure 3). 

### 3.4. Alpha-Diversity of AOM Communities

The alpha-diversities, which were measured as OTU richness, evenness, and Shannon index, were not significantly different between archaeal and bacterial *amo*A genes, but it revealed significant difference among wetland types. For the AOA *amoA* gene, the OTU richness in riverine wetlands was significantly higher than that in lacustrine wetlands, and the Shannon index in both riverine and palustrine wetlands was significantly higher than that in lacustrine wetlands (Figure 4). For the AOB *amoA* gene, both the OTU richness and the Shannon index in riverine wetlands were significantly higher than those in lacustrine and palustrine wetlands (Figure 4).

### 3.5. Spatial Distance and Environmental Variations Associated with Patterns of AOM Community Composition

The spatial distance in community composition estimated using beta-diversity based on Bray-Curtis distance, the Mantel test revealed significant distance-decay relationships (DDRs) of community similarity vs. geographic distance in both AOA (*r* = 0.201, *p* < 0.01) and AOB (*r* = 0.276, *p* < 0.01). Similarly, the abundant community similarities of AOA (*r* = 0.270, *p* < 0.01) and AOB (*r* = 0.401, *p* < 0.01) both generally declined with environmental variables (Figure 5).

Given that both spatial distance and environmental variation can be important in forming the AOM community, we attempted to explore their relative contribution. The variation partitioning analysis showed that the explained proportions of environmental factors (accounted for 66.86% and 68.18% variations in AOA and AOB, respectively) were dominantly higher than spatial variables (for less than 3% of the variations) and also indicted that the pure effect of environmental factors in the AOB community tended to be a bit stronger than in AOA community (Figure 6).

### 3.6. Environmental Drivers of AOM Abundance and Community Composition

The NMDS of taxonomic compositions of ammonia-oxidizing microbes did not clearly group by wetland types (top panels in Figure 7A), but rather separated by the local soil conductivities, as shown by the significant correlations between the first axis of NMDS and soil conductivities (bottom panels in Figure 7A). Based on the performance of local soil conductivity, we inferred that the environmental variables supposed to play important roles in AOM community structures. Accordingly, we further correlated distance-corrected dissimilarities of taxonomic community composition with environmental factors while using the partial Mantel test to identify the environmental drivers of AOM community structures in our research (Figure 7B). The AOA community composition were most strongly correlated with both MAT and MAP (Mantel’s *r* = 0.289–0.296; *p* < 0.01), while the soil pH, NO_3_^−^-N and plant richness were only weakly correlated with the AOA community (0.05 < *p* < 0.1) (Figure 7B). The AOB community assembly was strongly correlated with MAT, conductivity, and plant richness (Mantel’s *r* = 0.197–0.361, *p* < 0.01), and correlated with pH and TN (Mantel’s *r* = 0.127–0.225, *p* < 0.05). 

## 4. Discussion

### 4.1. Community Strucures of Ammonia-Oxidizers

In present study, we drew a comprehensive picture about the abundances, diversities, and compositions of AOA and AOB in high-elevation wetlands. We found the identities of ammonia-oxidizer sequences from Qinghai–Tibetan wetlands are similar with the AOM compositions from other studies. For example, *Nitrososphaera* clade (Group I.1b) overwhelmingly dominated (90.36%) in the AOA *amo*A sequences in our study, which was very close to the value (91.4%) from the study of *Nitrososphaera*-like microorganisms in cold springs on the Qinghai–Tibetan Plateau [35]. As for AOB sequences, the *Nitrosospira* clades were found to be more advantageous in our study. Similar results were also found in cold springs on the Tibetan plateau (*Nitrosospira* was account for 69.2% in AOB sequences) [35]. Previous studies reveal a phenomenon that *Nitrosospira*-like microbes was dominant in sediments from high-elevation habitats (Qinghai–Tibetan plateau, Yuan plateau, and Andean wetland) [35,36,37]. Besides, it is reported that the *Nitrosospira* group dominated at stations with a marine influence [38], which could explain why the local conductivity of soil and water has a prominent role in shaping AOB community.

### 4.2. Different Abundance but Similar α-Diversity Pattern of Ammonia-Oxidizers

The abundance differed substantially between the archaeal and bacterial *amoA* genes. Specifically, the abundance of AOA was higher in riverine and lacustrine wetlands, while AOB *amoA* gene dominated in palustrine wetlands. Generally, it is widely accepted that AOA prefer to survive in low NH_4_^+^ concentration [39,40,41,42]. In this study, the numerical advantage of AOB versus AOA in palustrine wetlands might be attributed to the higher soil NH_4_^+^ concentration (0.393~5.875 mg kg^−1^) in palustrine wetlands, as compared to riverine (NH_4_^+^: 0.201~1.121 mg kg^−1^) and lacustrine wetlands (NH_4_^+^: 0.247~1.708 mg kg^−1^). Particularly, these were two palustrine samples with the highest abundances of AOB in all samples, which also had the two highest NH_4_^+^ concentrations in all of the samples (Appendix A). Interestingly, numerous studies have reported that AOA are typically more abundant than AOB in natural wetlands, such as lake sediments [39,43,44] and marine environment [45,46,47,48], while AOB outnumber AOA was mainly observed in several high-elevation lakes [42,49,50] or artificial ecosystem, such as constructed wetlands [51], wastewater treatment plants [52], and bioreactors [53]. The findings of this study suggest that additional characteristics that are specific to high-elevation habitats may play an important role in shaping AOA and AOB community structure.

The copy numbers of AOA and AOB were detected in average of 2.63 × 10^4^ and 9.73 × 10^3^ copies g^−1^ in the wetlands of the Qinghai–Tibetan Plateau, prodigiously lower than those (1.34 × 10^5^ and 1.36 × 10^4^ copies g^−1^) in lakes in the Yangtze River basin of China, respectively [54,55]. A recent study reported AOA abundance values that ranged from 10^4^ to 10^9^ copies g^−1^ from aquatic ecosystems, including lakes, rivers, paddy fields, reservoirs, and swamps, from eight countries of all five continents [8]. The low temperature at the high-altitude conditions could account for this phenomenon. It has been suggested that AOA abundance was inversely correlated with elevation of lakes of the Sierra Nevada, USA [56]. Several studies also reported that the temperature appeared to be the crucial factor affecting the abundances of the ammonia-oxidizers in both natural and constructed wetlands [57,58,59].

When we separately analyzed both ammonia-oxidizers, there was almost no significant difference between every two types of wetland, except that the AOA abundance of riverine wetlands were much higher than those of other two types (the graphs on the top right of Figure 2). It is likely that the wetland type is not an important factor in determining the abundance of the ammonia-oxidizers in Qinghai-Tibetan Plateau. On the contrary, the alpha-diversities (measured as OTU richness, evenness and Shannon index) between each type of wetlands were different. In general, the diversities of AOA and AOB were lowest in lacustrine wetlands, and highest in riverine wetlands (Figure 4). This might be related to the salinity of the wetlands. Half of the lacustrine wetlands in this study are saline marshes and have an average water conductivity of 15,358 µS cm^−1^. The inference was supported by the negatively correlation between OTU richness, Shannon index, and the conductivity of water and soil (Appendix A). These findings suggested that conductivity largely contributed to the alpha-diversity of ammonia-oxidizers, which was in agreement with other researches. For instance, studies targeting the ammonia-oxidizing bacteria in the Chesapeake Bay [60] and MA estuaries [61] exhibited that the increasing salinity leads to the loss of diversity of AOB.

### 4.3. AOA and AOB Contributed Differently to Nitrification Activity

The ammonia-oxidizing microbes play a crucial role in wetland biogeochemical processes of wetland. We found that the nitrification was more related with AOB than AOA in our investigated sites. The relative contribution of AOA and AOB to the nitrification has been debated since Leininger et al. first reported the dominance of AOA in soil [62]. For instance, in a four-year repeated ammonium-fertilized study, the PNR significantly correlated with AOB abundance, but not with AOA abundance [63]. Taylor et al. also detected that AOB explained the majority of the nitrification activity in cropped soils, while, in fallowed soils, AOA and AOB both contributed to PNR [64]. The subsequent study illustrated that the soil conditions and NH_4_^+^-availability in soils might explain the different contributions of AOA and AOB to nitrification [65]. While in acidic soils, the changes in nitrification activities were reported driven by AOA abundance, but not AOB [66,67]. These contradictory results illustrated that AOA and AOB may differently contribute to the nitrification activity in heterogeneous environmental conditions.

### 4.4. Environmental Factors Shaping the Pattern of AOM Composition

The distance-decay relationship describes how the similarity in community assembly varies with the geographic distance that separates them, which has been widely used in spatial biodiversity studies [49,68,69]. In this study, a significant correlation was detected between community composition and spatial distance, and also between community composition and the environmental factor, which implies both geographic distance and environmental factors are important in forming the community assembly of AOM. However, the VPA results suggested that the environmental selection was likely to have a dominant role (Figure 6). Moreover, the environmental heterogeneity might strengthen the compositional variation over spaces, which tends to further enhance the distance-decay relationships (Appendix A).

Numerous studies have reported that oxygen concentration, salinity, pH, ammonia concentration, and temperature most likely affect the abundance of AOA [70,71,72] and AOB [42,43,73]. The above results were mainly obtained in a specific environment, and few studies have focused on a wider range [8]. The sites that we examined covered a broad spectrum of longitude (80.16~106.07°E), latitude (28.16~37.46°N), and elevation (254.4~5150.6 m), thus providing us with the opportunity to examine a broad range of environmental influences on AOM community composition. Our results suggest a dominant role for climate factors (MAT and MAP) in shaping community composition across a wide variety of wetland sites and conditions.

Although the environmental factors have the dominant role both in AOA and AOB community compositions, the relative importance and the driving factors are different. Our results indicated that the AOB communities in the Qinghai–Tibetan Plateau were more sensitive to changes in local environments than the AOA communities, which was consistent with other studies in Yangtze River [55] and in East Asian paddy soils [74]. Evolutionary considerations suggest that AOA can grow under the conditions of extreme salinity, temperature, and pH, and other conditions that are not suitable for the growth of AOB [43]. Moreover, the oligotrophic AOA could compete with bacteria for ammonia [11]. Therefore, AOA have a more versatile metabolism than AOB, and it might be adapted to a broader range of environments [62].

Clearly, these results highlight that the environmental factors have a prominent role in shaping the ammonia-oxidizer community patterns, but the drivers are different in AOA and AOB. It expands the ideas in wetland management that the characteristics of microorganism communities, especially those that are heavily involved in biogeochemical cycles, will serve as effective bioindicators to assess wetland trophic status and they will provide a potential application in the monitoring of wetlands [75,76].

## 5. Conclusions

The compositions and structures of AOA and AOB communities are of importance to the nitrification. However, less is known regarding the ammonia-oxidizer community in high-elevation wetland ecosystems. Here, we examined the AOM community compositions in a large-scale survey across the Qinghai-Tibetan Plateau. We found a strong heterogeneity in terms of the abundance, community, and activity of ammonia oxidizers of AOA and AOB in sediments from three types of wetland. The copy numbers of both archaeal and bacterial *amoA* genes were prodigiously lower in these high-elevation wetlands than detected in some previous studies in wetlands. AOA were the main ammonia oxidizers in riverine and lacustrine wetlands, while AOB dominated in palustrine wetlands. The net nitrification rate had a significant correlation to AOB, but not AOA abundance. The environmental conditions, but not spatial distance, have a dominant role in shaping the pattern of ammonia-oxidizer communities. Two climate factors (MAT and MAP) showed a strong impact on the total abundance of the archaeal *amoA* functional gene, while MAT and more soil conditions (conductivity and plant richness, pH and TN) were significantly correlated with the bacterial *amoA* functional gene. Further research should be conducted in order to reveal the role of complete nitrifiers (comammox) in the high-elevation wetlands, which were found as a novel *amo*A sequence group in recent years.

## Figures and Tables

**Figure 1 microorganisms-08-00933-f001:**
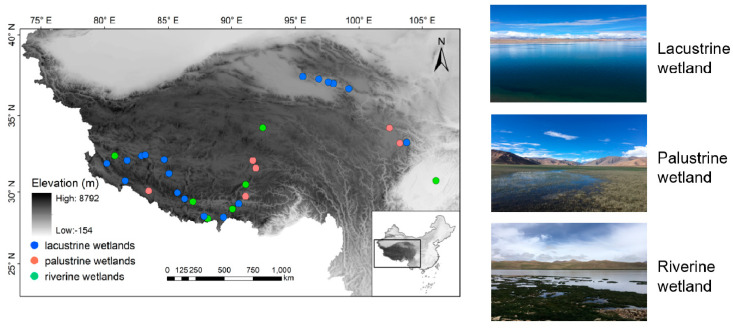
Location of the 31 wetlands on the Qinghai–Tibetan Plateau investigated in this study.

**Figure 2 microorganisms-08-00933-f002:**
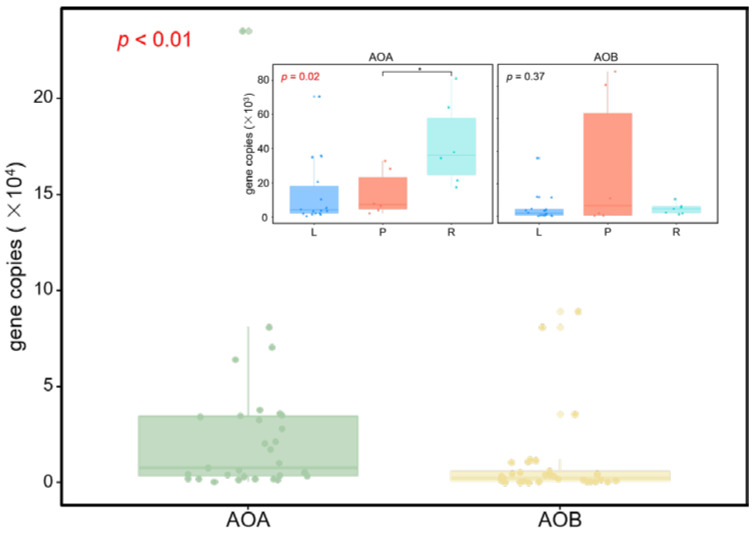
Abundance of ammonia-oxidizing microbe communities in soils. The graphs on the top right show the differences in abundances among lacustrine (L), palustrine (P), and riverine (R) wetlands. The asterisk indicates the statistical significance between two wetland types (* *p* < 0.05).

**Figure 3 microorganisms-08-00933-f003:**
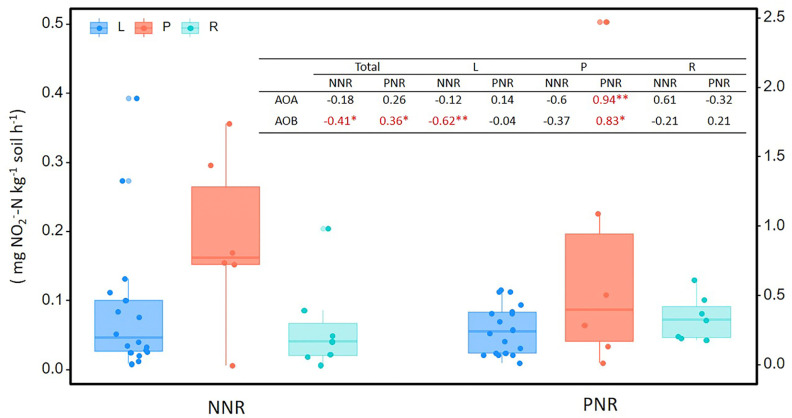
The net nitrification rate (NNR) and the potential nitrification rate (PNR) in soils from lacustrine (L), palustrine (P), and riverine (R) wetlands. The table shows the Spearman’s r and the statistical significance of the ammonia-oxidizers abundances correlated to NNR and PNR in total and different wetlands (** *p* < 0.01, * *p* < 0.05).

**Figure 4 microorganisms-08-00933-f004:**
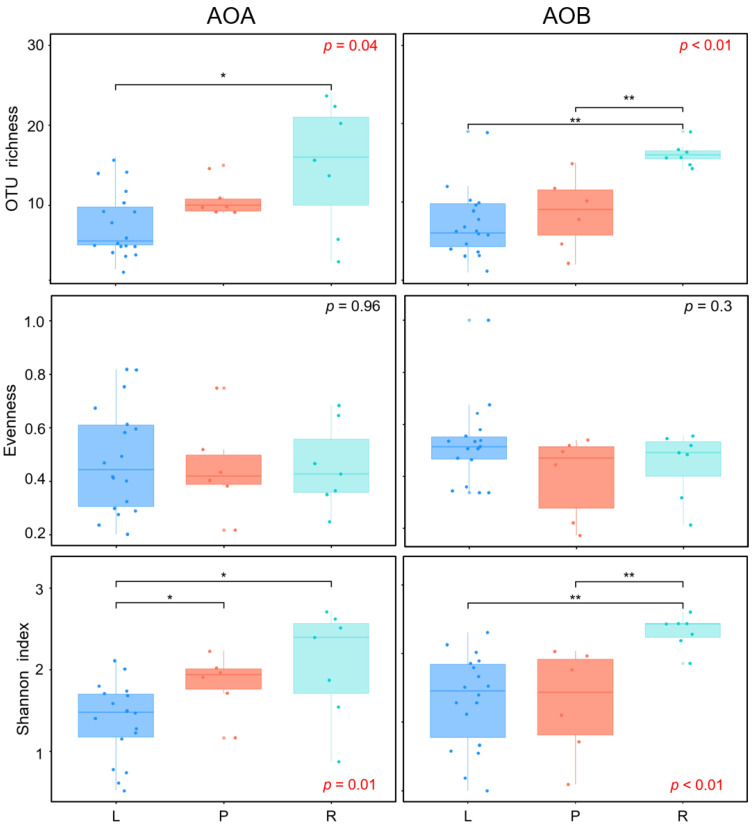
Alpha diversity of ammonia-oxidizing microbe communities in soils collected from lacustrine (L), palustrine (P), and riverine (R) wetlands. The asterisk indicates the statistically significant between two wetland types (** *p* < 0.01, * *p* < 0.05).

**Figure 5 microorganisms-08-00933-f005:**
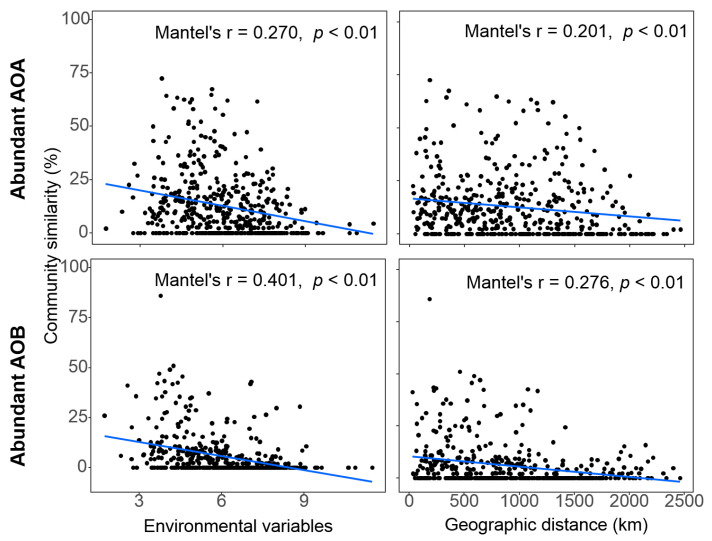
Spearman’s rank correlation between ammonia-oxidizing microbe communities (Bray-Curtis similarity) and the Euclidean distance of environmental factors, geographic distance. The blue lines denote the linear regression across all sampling sites. The Mantel test was used to examine the correlations between the pairwise Bray-Curtis similarity and pairwise differences in environmental variables/geographic distance. All 19 environmental variables were used.

**Figure 6 microorganisms-08-00933-f006:**
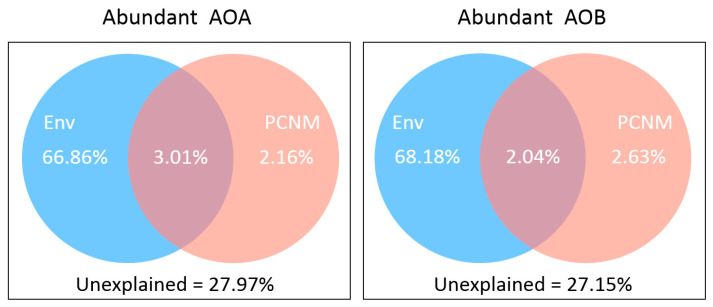
Variation partitioning of ammonia-oxidizing microbe communities by environmental variables (Env) and spatial factors (PCNM) for abundant communities.

**Figure 7 microorganisms-08-00933-f007:**
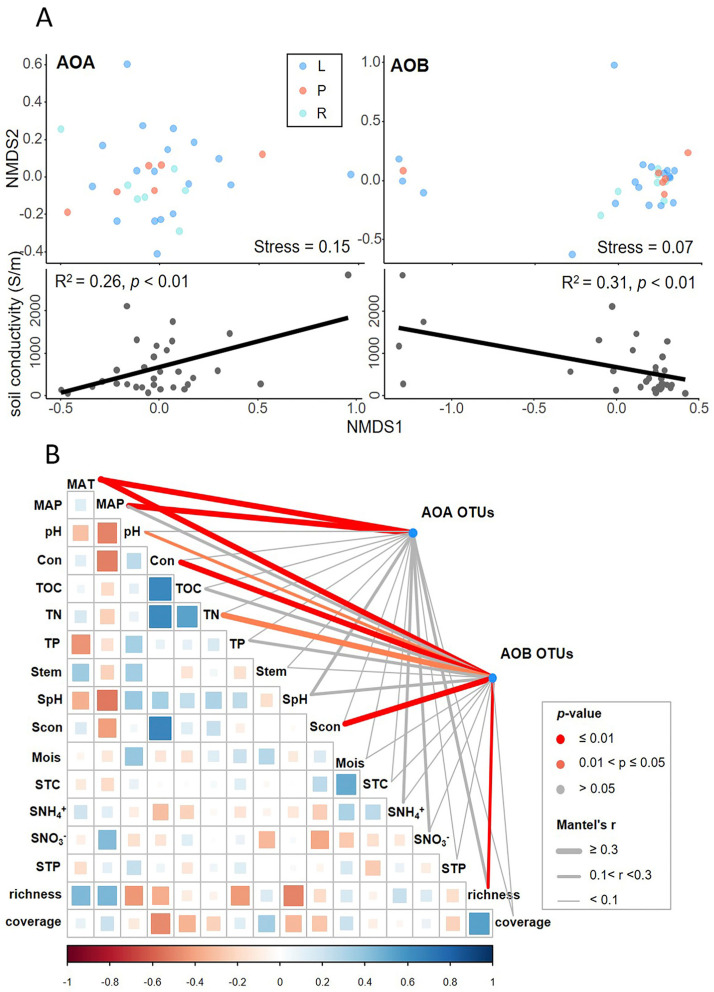
Environmental drivers of ammonia-oxidizing microbe community compositions. (**A**) Non-metric multidimensional scaling (NMDS) ordination based on unweighted Unifrac distances similarity shows that ammonia-oxidizing microbe community composition are not grouped clearly by wetland types (top), but rather separated by the local soil conductivities as shown by the significant correlations (bottom). (**B**) Pairwise comparisons of environmental factors are shown, with a color gradient denoting Spearman’s rank correlation coefficient. Ammonia-oxidizing archaea (AOA) and ammonia-oxidizing bacteria (AOB) community composition (Bray-Curtis distance) was related to each environmental factor by partial (geographic distance corrected) Mantel test. Edge width denotes the Mantel’s r statistic and the edge color corresponds to the statistical significance based on 9999 permutations. MAT: mean annual temperature; MAP: mean annual precipitation; Con: conductivity in water; TOC: total organic carbon in water; TN: total nitrogen in water; TP: total phosphorous in water; Stem: soil temperature; SpH: soil pH; Scon: conductivity in soil; Mois: soil moisture; STC: total carbon in soil; SNH_4_^+^: soil NH_4_^+^-N; SNO_3_^−^: soil NO_3_^−^-N; STP: total phosphorous in soil; richness: plant species richness; coverage: plant coverage.

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
