# Peer review of "Environmental Factors, More Than Spatial Distance, Explain Community Structure of Soil Ammonia-Oxidizers in Wetlands on the Qinghai–Tibetan Plateau"

_microorganisms, 2020, doi:10.3390/microorganisms8060933_

Round 1

Reviewer 1 Report

In this study, the authors analyzed the ammonia-oxidizing archaea and bacteria abundance and diversity in sediments samples of 3 different types of wetlands at the Qinghai-Tibetan Plateau. qPCRs were performed to quantify amoA (the alpha subunit of ammonia monooxygenase, key enzyme of ammonia oxidation) copies in the different samples and diversity was analyzed using clone libraries (60 clones per site). In addition, net and potential nitrification rates were determined.

Several studies comparing ammonia-oxidizing archaea (AOA) and ammonia-oxidising bacteria (AOB) abundance and diversities have been already conducted analyzing numerous habitats. The novel aspect of this study is that in total 31 sampling sites were analyzed across the Qinghai-Tibean Plateau. While the number of sampling sites is impressive, the study lack substantial details in the description of each sampling site and results per sampling site (see comment about number of positive amoA clones and Online resource 1). Especially, since the study mainly focusses on the diversity of AOB and AOA communities, the exact number of amoA-positive clones, OTU, and unique OTU per sampling site should be given. Also, rarefraction curves should be shown to show whether the majority of diversity was already covered by sampling.

In addition, insights into the composition of AOA and AOB community (see comment L21-22, 51 and L72-75) are lacking. So the (dis)similarities of the communities are described/discussed but not “who” is there, which is an essential question in every sequencing-based microbial ecology study.

Specific comments:

L21-22, 51 and L72-75: While abundance and diversity of AOA vs AOB were described, the community composition/structure of AOA and AOB were not analyzed by the authors, which is one of my major criticism of the study, since a phylogenetic analysis of the obtained sequences to get insights into the community composition of the different ammonia-oxidizing guilds is golden standard and often performed in similar studies (e.g. citation in the manuscript: 10, 15 or https://www.frontiersin.org/articles/10.3389/fmicb.2014.00743/full (similar study)).

L27-29: “The AOA abundance was influenced by mean annual temperature (MAT) and mean annual precipitation (MAP), while the AOA abundance was influenced by MAT, conductivity and plant richness, pH and TN.“ Is it the abundance or the community composition (thus the OTU diversity)?

L50-51: “We do not know if high-elevation wetlands have bacterial aspects similar to other wetlands“ Only bacterial aspects? Or should this statement should include AOA?

L52-56: These two previous studies on ammonia oxidizers in river sediments in the Qinghai-Tibetan Plateau analyzed the distribution of complete nitrifiers (comammox), the third known group to aerobically oxidize ammonia. In addition, other studies, detected comammox bacteria in wetlands (e.g.  Thus, why is this study only focused on AOA and AOB, without even mention comammox in the text (explaining why this study excluded comammox from all analyses), or doing a PCR to show the presence/absence of comammox bacteria in the wetland sediments?

L80: Although the average elevation was higher than 4 000m, the range was very high from 254 – 5151; Does this range of elevation also influenced AOB/AOA diversity? Could this pronounced elevation range in the riverine wetlands (254.4~4545.3) explain why the richness for AOB and AOA was higher in this wetland type?

L148-149: “Approximately sixty positive clones with correct size from each site were selected for sequencing.” Since the number of positive clones per site per ammonia-oxidizing clade is a relevant information, the exact numbers should be given in a table (e.g in online resource). This table should include per sampling site: number of positive clones (AOA), number of positive clones (AOB), number of OTU (AOA), number of OTU (AOB), number of unique OTU (AOA), number of unique OTU (AOB).

L172-174; L216-217: Definition of regionally rare OTUs: < 0.01% relative abundance in local sample means less than 1 sequence in 10 000. Since based on L148 “approximately sixty positive clones with correct size from each site” were selected/sequenced, it is not possible to find rare OTUs ( I guess this definition comes from amplicon studies; also mean relative abundance in local samples, is not possible because there are no replicates per site).

L290 – 291: change NH3 to NH4+ concentration as according to the online resource 1, this is NH4+.

L290 – 291: Based on Figure 2 there are two samples in the boxplot that show high AOB abundance; are these the one that have a high NH4+ concentration and low pH? The range of both parameters is so large in this wetland types that without this info, it is difficult to hypothesize low pH and high NH4+ might cause this abundance.

L319-320 “These findings suggested that conductivity contributed largely to the alpha-diversity of ammonia-oxidizers.“ -> Did other studies come to similar conclusions?

L357: include citation in citation list : Wang et al 2019

L367-368: phytoplankton in wetland sediment?

Figure 4: L240: “The net nitrification rate (NNR) and the potential nitrification rate (PNR) in dry soils from lacustrine (L), palustrine (P) and riverine (R) wetlands.” -> NNR and PNR were determined using soil slurry, not dry soil based on material and methods section.

L378: please rephrase: in three types of wetland sediments.

Figures:

In most plots comparing AOA and AOB the axis of the AOA and AOB plot differ, which makes it hard to compare the values.

Figure 1: The 7th riverine wetland sampling site is not visible.

Figure 7 and online resource 1 and 4: Does the variable species richness describe plant species richness? If so then this is not clear.

Online material:

Online resource 1:

To summarize the sampling sites per wetland types makes sense in order to keep the list relatively short. However, a full list that include the geographic, climatic, physicochemical properties, and botanic and AOM variables for every site should be also included.

Online resource 2:

Why is the reference of the qPCR this study? Also others performed already qPCR with this primer set.

Reviewer 2 Report

The manuscript "Environmental Factors, More Than Spatial Distance, Explain Community Sturcture of Soil Ammonia-Oxidizers in Wetland on the Qinghai-Tibetan Plateau" analyzes the environmental factors that shape the microbial ammonia-oxidizing communities in three different types of wetland soils across the Qinghai-Tibetan Plateau. Clone libraries and qPCR assays are applied and correlated with environmental data. Methods and results are sound. Yet, the study could benefit from re-structuring the text and setting the results better into literature context.

Comments on Figures:

  • Fig. 2 & Fig. 3: use the same y-axis labeling for AOA and AOB to make them easier to compare.
  • Fig. 3: explain what two asterisks mean (same as in Fig. 4?)
  • Fig. 5: What is the message of this figure? The blue lines through these point clouds look rather random and the legend is not clear. There is also a mix-up in the legend - I don’t see (A) and (B) in the figure and the order is not right. And there is a (1 – Bray-Curtis dissimilarity) without a (2 - ??). If the p value is the same for all four panels, it could be given in the legend and removed from within the figure.
  • Fig. 6: Is this the right legend? (A) and (B) is not indicated in the figure and the text does not correspond with the figure content.
  • Fig. 7 A is not explained well: How do the lower and upper panels correspond? Why is soil conductivity shown here and no other factors? If I am reading panel B right, it is an important factor for AOB communities, but not for AOA.

Comments by line numbers:

  • 43: relative abundances
  • 48-50: so other studies have shown the importance of the same factors before? For which environments? It is not clear from this sentence, what the novelty of this study is. Also references are missing for the first part of the sentence.
  • 57-58: ammonia-oxidizer communities
  • 71-72: A temperature range of ca. 20°C is not very extreme – does “highly heterogeneous” apply? Maybe temperature is not the best factor to show the heterogeneity of these environments.
  • 84: environments
  • 85: remove “which”
  • 85-86: “developed by Tammi” – who is that? Reference?
  • 92-93: What does that mean? State clearly how these samples were stored.
  • 101: Plant coverage
  • 110: reword: “a stainless probe of thermometer”
  • 112: conductivity
  • 121: 5 g… were weighted
  • 126: “after centrifuged”: were they filtered first or centrifuged first? Either: “filtered through … filters and then centrifuged at…” or “filgered through… filters after centrifugation at…”
  • 129-131: re-phrase
  • 135: manufacturer’s
  • 139-140: Primer sequences and … are shown in Online Resource 2.
  • 151-152: were aligned using MAFFT
  • 152-153: Sequences sharing…using the program Mothur with the furthest neighbor algorithm
  • 158: unclear: “of Arch-amoA and amoA genes fragments”
  • 161: are listed in
  • 162: colonies
  • 162-163: which was confirmed that… ?? There is something off with this sentence.
  • 170 in recent literature/studies/publications
  • 177-188: provide references for software
  • 185: non-metric
  • 207: wetlands
  • 209-210: However, in palustrine wetlands AOB (…) were significantly more abundant than AOA (…).
  • 217: remove “Venn diagram analysis showed”
  • 217-219: this sentence is unclear. Very complicated sentence. Please check the figure and re-phrase. What is the message here?
  • 220-225: I feel, the “flight level” of the text is jumping back and forward – the paragraph before was already at OTU-level. Now it is alpha diversity, which is more general. Please structure the text in a consistent way from more general to more specific aspects. The text is rather hard to read – instead of describing the actual index values, it would be more helpful to provide an interpretation, e.g. “Shannon index and OTU richness showed that riverine and palustrine wetlands had significantly more diverse AOA communities than lacustrine wetlands”
  • 231-238: Again, this text is just describing the figure without interpretation. Therefore, this text and Fig. 4 are redundant.
  • 254: no panel (C) is indicated in Figure 5.
  • There is no reference to Figure 6 in the text.
  • 263-265: I see the separation into two groups for AOB and also the correlation with soil conductivity for AOB, but not for AOA. Why is MAT not shown in Fig. 7A instead of soil conductivity (or additional), as it is significant for both AOA and AOB? Or does that not apply to the NMDS analysis? There is a lot of information shown in Figure 7, but little explanation.
  • Discussion: very abrupt start – It would be good to provide the literature background first; Providing the measured environmental factors (pH, nitrogen etc.) here is too late – it needs to be somewhere in the results.
  • 297-299: “These findings” – does that mean “The findings of this study”? – very generic sentence.
  • 300-302: This is a very specific comparison to values from one lake – why?
  • Discussion in general: this feels like a collection of references that somehow relate to this study. Rather specific sentences on the data in this study are alternating with general findings of other studies - but the connection and interpretation is missing. Parts of the text would be more fitting in the results part (e.g., l. 323-327; l. 341-354)
  • Comparisons are only made to other Asian environments. What is the literature base on other parts of the world?
  • Conclusions: this is just a summary of the data in this study – there is no context.

Round 2

Reviewer 2 Report

All reviewer comments were addressed appropriately and I find the manuscript highly improved. 

Minor comments:

-l.167: p missing in replicates

-l.250: "Nitrosopurmilus" should likely be "Nitrosopumilus"

-Fig.4: The panels seem to not have the same hight - check especially the top two AOB panels.

-l.310: "group"

-l.346: "Nitriososphaera-like" should probably be "Nitrososphaera-like"

-l.350: "was dominant"

-l.444: "relative"; and "driving factors"

Author Response

We thank the reviewer for the helpful comments, which have addressed as follows.

-l.167: p missing in replicates

Response—“relicates” has been amended to “replicates”.

-l.250: "Nitrosopurmilus" should likely be "Nitrosopumilus"

Response—“Nitrosopurmilus” has been amended to “Nitrosopumilus”.

-Fig.4: The panels seem to not have the same hight - check especially the top two AOB panels.

Response—Fig. 4 has been redrawn with the same panel height.

-l.310: "group"

Response—“grouped” has been amended to “group”.

-l.346: "Nitriososphaera-like" should probably be "Nitrososphaera-like"

Response—“Nitriososphaera-like” has been amended to “Nitrososphaera-like”.

-l.350: "was dominant"

Response—“was dominate” has been amended to “was dominant”.

-l.444: "relative"; and "driving factors"

Response—“reletive” has been amended to “relative”, and “driven factors” has been amended to “driving factors”.